# Optimizing Straw-Rotting Cultivation for Sustainable Edible Mushroom Production: Composting Spent Mushroom Substrate with Straw Additions

**DOI:** 10.3390/jof9090925

**Published:** 2023-09-13

**Authors:** Yongsheng Ma, Lingyun Liu, Xiaoyan Zhou, Tian Tian, Shuai Xu, Dan Li, Changtian Li, Yu Li

**Affiliations:** 1Engineering Research Center of Edible and Medicinal Fungi, Ministry of Education, Jilin Agricultural University, Changchun 130118, China; mayongsheng5303@163.com (Y.M.); liulingyun0315@163.com (L.L.); zxy@jlau.edu.cn (X.Z.); tiantian9437@126.com (T.T.); xushuai@jlau.edu.cn (S.X.); junwuzhongxin@126.com (D.L.); yuli966@126.com (Y.L.); 2International Joint Research Center for the Creation of New Edible Mushroom Germplasm Resources, Ministry of Science and Technology, Jilin Agricultural University, Changchun 130118, China

**Keywords:** SMS, biological pretreatment, microbial diversity, total humus carbon, composting

## Abstract

In recent years, the optimization of straw-rotting formulations for cultivating edible mushrooms and the management of the resulting spent mushroom substrate have emerged as new challenges. This study aimed to investigate the composting of spent mushroom substrate produced from mushroom cultivation with various straw additions, under conditions where chicken manure was also used. Parameters measured during the composting process included temperature, pH, electrical conductivity (EC), germination index (GI), moisture, and total nitrogen content. Additionally, changes in nutrient content within the compost piles before and after composting were determined, and the variations in bacterial and fungal communities across different treatments before and after composting were analyzed using 16S rRNA and ITS sequencing. The results indicated that the spent mushroom substrate produced by adding 20% straw during mushroom cultivation was more suitable for composting treatment. The findings suggest that incorporating an appropriate amount of straw in mushroom cultivation can facilitate subsequent composting of spent mushroom substrate, providing an effective strategy for both environmental protection and cost reduction.

## 1. Introduction

Since 1978, global edible mushroom production has exhibited significant growth, increasing more than 30-fold [1]. The per capita consumption of mushrooms continues to rise. China, as a major producer of edible mushrooms [2], ranks its mushroom industry fifth among agricultural products, following grains and oils, fruits, and vegetables [3,4].

Considering the varying nutrient requirements of different species of wood-rotting edible fungi, corn stover can partially or entirely substitute wood chips, providing a low-cost, stable, and high-quality cultivation substrate. Therefore, using straw for cultivating edible mushrooms with a multi-stage recycling model plays a crucial role in enhancing economic and social benefits. However, with the yearly increase in straw-cultivated edible mushroom production, disposing of the resulting residue presents a challenge. The spent mushroom substrate (SMS) is a lignocellulosic byproduct generated during mushroom cultivation. Since the degradation efficiency of edible mushrooms is only 40–80% [5], the lignocellulose in SMS cannot be entirely degraded. SMS primarily consists of nutrients, residual fungal mycelium, various decomposing lignocellulosic biomass, and high concentrations of organic matter and other substances. Applying SMS to agroecosystems is considered a sustainable solution for effectively utilizing this product, including as an alternative to fertilizers and soil conditioners [6]. However, potential hazards, such as the presence of disease-causing microorganisms and pathogenic bacteria, may arise if spent mushroom substrate is applied to soil without proper composting treatment. To circumvent these potential issues, spent mushroom substrate can undergo appropriate composting.

Composting technology has garnered considerable attention as an environmentally friendly waste treatment method [7]. The high temperature during composting can eliminate disease-causing microorganisms and pathogenic bacteria in the spent mushroom substrate, rendering it more suitable for plant growth. Composted mushroom residue can serve as a high-quality organic fertilizer, supplying ample nutrients and a favorable growth environment for crops. Unlike traditional single-material composting, co-composting is a technique that processes multiple materials simultaneously, promoting organic matter decomposition and accelerating the composting process [8]. For instance, chicken manure contains high levels of nitrogen, phosphorus, and potassium, whereas mushroom residue has high carbon and cellulose content. Mixing the two balances the nutrients, making the compost more balanced and nutrient-rich, thereby achieving optimal treatment results [9]. Co-composting can shorten the fermentation period, expedite the humification process, and enhance maturity compared to traditional composting methods [10].

This study aims to explore the effects of aerobic composting of spent mushroom substrate generated from mushroom cultivation using chicken manure and different proportions of straw additions. By measuring the physical, chemical, and microbial changes during the combined composting process, the influence of straw addition on composting after mushroom cultivation is analyzed to determine the most suitable straw addition amount. The purpose of this research is to provide a more environmentally friendly and efficient waste treatment method for the mushroom industry, thereby reducing the industry’s impact on the environment and increasing resource recycling rates.

## 2. Material and Methods

### 2.1. Composting Materials and Experimental Design

The composting materials used in this study consisted of chicken manure and *Auricularia cornea* cv. Yu Muer residue, along with straw. The chicken manure was sourced from Jilin Agricultural University, while the Yu Muer residue was obtained from the university’s mushroom and vegetable base. Table 1 illustrates the formulation of the Yu Muer residue combined with added straw. The primary components of the inoculum include *Bacillus subtilis*, *Bacillus licheniformis*, and *Bacillus laterosporus*.

The composting experiment took place at the mushroom and vegetable base of Jilin Agricultural University in Changchun, China. The reactors comprised fifteen insulated foam boxes, each measuring 57 cm in length, 42.5 cm in width, 30 cm in height, and 27 mm in thickness. After determining the moisture content of the Yu Muer residue and chicken manure combined with straw, the materials were mixed at a dry weight ratio of 3:1 and adjusted to achieve a 60% moisture content for the compost. Each treated compost mixture weighed 15 kg. A bacteriological agent, primarily composed of *Bacillus* sp., was added at a rate of 1% of the total dry weight.

### 2.2. On-Site Sampling and Monitoring

The fertilization process, lasting 30 days, involved turning the piles on days 0, 3, 6, 10, 17, 23, and 30 to promote aeration. Samples were collected from the top, middle, and bottom sections of each reactor. Once the samples were thoroughly mixed, they were divided into three parts. The first part was air-dried, crushed, and passed through a 0.1 mm soil sieve at room temperature for the determination of organic matter, total nitrogen, phosphorus, and potassium. The second part was stored in a refrigerator at 4 °C for analyzing physical and chemical parameters, as well as NH^4+^, NO^3−^, and humic substances. The third part was preserved at −80 °C for assessing microbial diversity.

### 2.3. Analysis Methods

#### 2.3.1. Analysis of Physical and Chemical Parameters

The compost core temperature was monitored three times daily (8:00, 14:00, and 20:00). Fresh sample moisture content was determined by heating the samples at 105 °C for 20 min using an infrared moisture analyzer. Seed germination toxicity was assessed in the novel compost formulations using cabbage seeds. The new formulations were mixed with deionized water at a 1:10 (volume-to-weight) ratio and placed in a shaker. The samples were shaken horizontally for 1 h at 25 °C and 180 r/min. The extracts were then centrifuged at 25 °C and 4500 r/min for 10 min, and the supernatant was collected. pH and conductivity measurements were taken using a pH meter and conductivity meter, respectively. Next, 10 mL of the extract was added to a 9 cm diameter Petri dish lined with two layers of filter paper, and 20 cabbage seeds were evenly distributed in each dish using tweezers. Deionized water served as the control treatment, following the same procedure. Each treatment, including the control, was performed in triplicate. The prepared Petri dishes were placed in a constant-temperature incubator at 25 °C for 72 h. The seed germination index was calculated using the following equation:GI(%) = (R × L)/(R_0_ × L_0_) × 100%
where GI is the seed germination index (%), R is the seed germination rate (%) of compost extract, L is the average root length (mm) of compost extract, R_0_ is the seed germination rate (%) of deionized water, and L_0_ is the average root length (mm) of deionized water seeds.

#### 2.3.2. Determination of Physical and Chemical Indicators

The total nitrogen content was measured using the Kjeldahl method following digestion with sulfuric acid. Ammonium nitrogen content was determined by extracting with a potassium chloride solution and employing indophenol blue colorimetry. The nitrate nitrogen content was assessed using the double-wavelength colorimetric method, which involved leaching with a potassium chloride solution. To determine the available phosphorus, sodium bicarbonate/sodium fluoride hydrochloric acid extraction was used, followed by the molybdenum–antimony colorimetric method. Available potassium was measured using ammonium acetate extraction and a flame photometer. The organic matter (organic carbon) content was determined using the potassium dichromate volumetric method (external heating method). Humus content was assessed using the potassium dichromate oxidation method after extraction with sodium pyrophosphate and sodium hydroxide. Soluble humus was extracted using a 0.1 mol/L sodium pyrophosphate sodium hydroxide mixture, and the total amounts of humic acid and fulvic acid were determined using the potassium dichromate oxidation volumetric method. The cellulose, hemicellulose, and lignin content were measured using the Solarbio kit according to the manufacturer’s instructions.

#### 2.3.3. DNA Extraction and High-Throughput Sequencing

To investigate the changes in microbial communities during the composting process, total DNA from the microbiome was extracted from various samples using the CTAB method. DNA mass and concentration were determined through agarose gel electrophoresis and ultraviolet spectrophotometry. PCR amplification targeted the 16S rDNA (V3–V4) region of bacteria and the ITS2 region of fungi as conserved microbial diversity DNA regions, utilizing primers from Shanghai Biotree Biotech Co. Ltd. The universal primers for bacteria were 341F (5’-CCTACGGGNGGWGCAG-3’) and 805R (5’-GACTACHVGGGTATCTAATCC-3’); for fungi, the primers were ITS1FI2 (5’-GTGARTCATCGAATCTTTG-3’) and ITS2 (5’-TCCTCCCTTATTGC-3’). Subsequent steps included PCR product quantification, amplification product recovery, and purification. The purified PCR products were evaluated using an Agilent 2100 bioanalyzer (Agilent, USA) and Illumina (Kapa Biosciences, Woburn, MA, USA) library quantification kits to generate sequencing libraries. The qualified up-sequencing libraries (with non-repeating index sequences) were mixed in the appropriate ratios according to the required sequencing volume and denatured to a single strand with NaOH. The NovaSeq 6000 sequencer performed 2 × 250 bp paired-end sequencing using the NovaSeq 6000 SP Reagent Kit (500 cycles).

#### 2.3.4. Microbial Data Analysis and Processing

The paired-end sequencing data were demultiplexed based on barcode information, and the splice and barcode sequences were removed. Next, the DADA2 algorithm was employed for length filtering and denoising using the command, generating the ASV (feature) sequence and ASV (feature) abundance table. Alpha and beta diversity analyses were conducted using the obtained ASV (feature) sequence and ASV (feature) abundance table. Bacterial species annotation was performed using the SILVA and NT-16S databases based on the ASV (feature) sequence file, while the RDP database was used for Fungal species via the annotation.qiime dada2 denoise-paired protocol. The abundance of each species in the ASV (feature) abundance table was determined for each sample, with a confidence threshold set at 0.7. Statistical analyses were performed to examine differences between comparison groups based on species richness information. Different statistical methods were applied depending on the sample conditions. For instance, Fisher’s exact test was used for comparing differences between samples without biological replicates; the Mann–Whitney U test was employed for comparing differences between two groups of samples with biological replicates; and the Kruskal–Wallis test was utilized for comparing differences between multiple groups of samples with biological replicates. The screening threshold for significance was set at *p* < 0.05.

### 2.4. Statistical Analysis

Microsoft Excel 2017 software was used for data processing and analysis. The two-factor analysis of variance (ANOVA) in SPSS 22.0 software tests was used to test for significant differences. Differences were considered significant at a level of *p* < 0.05. The data are represented as mean ± standard deviation (SD).

## 3. Results and Discussion

### 3.1. Changes during the Composting Process

During the composting process, real-time temperature measurement is essential for monitoring composting progress and microbial activity, as temperature serves as a sensitive indicator of composting stages [11]. Aerobic high-temperature composting typically involves three stages known as the warming, high-temperature, and cooling stages. The high-temperature stage is critical for eliminating pathogenic bacteria, toxic substances, and rapidly decomposing organic matter [12]. In this study, composting lasted for 30 days, with temperature measurements taken at 8:00, 14:00, and 20:00, and subsequent temperature profiles plotted (Figure 1). The temperature of all four treatment groups reached the highest level (>50 °C) on the second day, entering the high-temperature stage due to the addition of microbial agents that accelerated the composting reaction and increased the compost’s temperature [13]. Among the treatments, A3 warmed up significantly, reaching 62.67 °C the following morning at 8:00, likely due to the high content of easily decomposable organic matter, such as cellulose and hemicellulose, in the straw [14]. All four treatments (CK, A1, A2, and A3) maintained temperatures above 55 °C for more than three days, satisfying the requirement for eradicating disease-causing microorganisms and ensuring that the compost’s sanitary index was adequate [15]. Treatment A1 had the longest duration in the high-temperature phase, indicating better performance in eliminating harmful organisms and improving the compost’s biological stability through continuous high temperature [16].

Moisture content is a critical factor influencing microbial activity [17]. In this study, the moisture content of all four treatments was uniformly adjusted to 60%, and the four treatments exhibited similar moisture content at the end of composting. pH serves as a vital indicator for evaluating microbial activity during composting, and for optimal results, the pH should be maintained between 6.0 and 9.0, as deviations may adversely affect the composting process [18]. From Figure 2b, we can see that in the first three days of this study, the rate of pH increase was the fastest in each treatment, likely due to the high organic mass at the beginning of the composting period, which promoted microbial activity and volatilization of small molecule organic acids with rising temperature [19]. Throughout the composting process, the pH of the A3 treatment consistently remained higher than that of the other treatments after the third day. Furthermore, from the final pH measurements, it was observed that pH increased with the rise in straw content in the treatments. The final pile products of CK and A1 were slightly alkaline, which significantly differed from the other two treatments and were more suitable for seedling growth [20].

Electrical conductivity (EC) serves as an indicator of the compost substrate’s salinity. As substrate degradation progresses, EC tends to increase, and when it surpasses 4.0 ms/cm, it can adversely impact plant growth [21]. In this study, the EC values of each treatment gradually rose over time (Figure 2c), particularly during the late stage of composting, which may be attributed to the net loss of water due to organic matter decomposition and evaporation losses [22]. Notably, the A3 treatment attained a conductivity of 4.43 ms/cm at the end of composting, which significantly differed from the other treatments. This was likely due to the higher quantity of straw added to the A3 treatment compared to the other treatments, and the final conductivity of the A3 treatment surpassed that of the straw composting conducted by Xingyao Meng et al. [23]. The final conductivity of the A3 treatment exceeded the safe threshold for plant growth, rendering it unsuitable for further testing. However, the remaining three treatments exhibited conductivity values below 4.0 ms/cm at the end of composting, indicating no adverse effects on plant growth.

The seed germination index (GI) serves as a vital indicator for assessing compost maturity. Previous research has demonstrated that incorporating bacterial residues into organic waste compost can elevate the GI value and diminish the phytotoxicity of the final product [24]. In this study, the initial GI values for each treatment were high, with some surpassing 80% (Figure 2d). Notably, the A3 treatment significantly differed from the other treatments, exhibiting a high seed germination rate of 123.2% at the beginning of composting. This was likely due to the excessive amount of straw in this formulation but also suggested that the A3 treatment positively influenced seed germination during the initial composting stage. Similar to Naser Khan’s study, which revealed that compost produced from chicken manure and sawdust had low phytotoxicity even at day 0 and throughout the high-temperature stage of composting [25], a comparable phenomenon may have taken place in the current investigation. Although the GI values of the A3 treatment were initially high, they fluctuated during the composting process. However, upon the completion of composting, the GI values of the A3 treatment did not vary significantly from their initial levels. The seed germination of the CK treatment progressively increased throughout the composting process, whereas the A1, A2, and A3 treatments displayed a decline in seed germination during the first three days, possibly due to the release of high concentrations of ammonia and low molecular weight organic acids, which led to a decrease in GI [26]. The GI of these treatments then gradually rose, particularly in the A1 treatment, where GI values began to rapidly increase on the third day after reaching their lowest value and subsequently peaked as the highest value among all treatments by the tenth day. At the end of the compost pile, the A1 treatment exhibited the highest seed germination rate of 142.3%, significantly differing from the other treatments. The superior performance of the A1 treatment, in comparison to other treatments, may be attributed to the addition of 20% straw, which optimized the carbon-to-nitrogen ratio for composting treatment, or due to the appropriate quantity of straw enhancing the pile’s permeability, thereby facilitating microbial growth and promoting compost maturation. This suggests that adding the correct amount of straw can foster pile maturation.

### 3.2. Degradation of Lignocellulose

Organic solid waste contains significant quantities of recalcitrant organic carbon components such as cellulose, hemicellulose, and lignin. Cellulose and hemicellulose consist of various types of polysaccharides, while lignin is an aromatic polymer with a three-dimensional network structure. These components are tightly intertwined and exhibit high resistance to degradation [27]. Composting enables microorganisms to degrade lignocellulose and convert it into humus, thus transforming organic solid waste into organic fertilizer and soil conditioner [28]. In this experiment, a Bacillus-subtilis-based bacterium was introduced to each pile to augment the population of functional microorganisms, enhance the biodegradation of organic matter, and improve compost maturity [29]. Fungi possess a competitive advantage over bacteria in lignocellulosic degradation processes due to their mycelial structure. Ascomycota and Basidiomycota are the predominant fungal phylum in lignocellulose degradation, and their high abundance facilitates the degradation of organic waste during composting. *Chaetomium*, *Penicillium*, Trichoderma, *Arthrographis*, and *Aspergillus* are the core fungal genera in the composting process and can degrade lignocellulose by secreting lignocellulases during the composting process [30,31]. Lignocellulases can be classified into cellulases, hemicellulases, and ligninases according to the nature of the substrate, and they are involved in the degradation of cellulose, hemicellulose, and lignin in the composting process. These fungal genus were monitored in the piles throughout the subsequent fungal fraction analysis in this study. The experimental results revealed that cellulose degradation occurred most rapidly in the four treatments during the initial composting phase, indicating increased activity of composting microbiota during the high-temperature period of composting (Figure 3a). This is because cellulose decomposition yields polysaccharides and monosaccharides, which provide energy for microbial metabolic activity [32]. By the end of composting, 65.4%, 75.6%, 78.6%, and 84.5% of cellulose was degraded in the CK, A1, A2, and A3 treatments, respectively, suggesting that cellulose degradation increased with the quantity of straw added to the treatment. Hemicellulose and lignin demonstrated a gradual decrease throughout the composting process (Figure 3a,b). However, the hemicellulose and lignin contents of the A3 treatment were lower than those of the other treatments, potentially due to the greater amount of straw added in this treatment.

### 3.3. Changes in Nutrient Content

The nutrient content of compost serves as a crucial indicator of its quality. In this study, we evaluated compost nutrient content by measuring the levels of readily available phosphorus, readily available potassium, nitrate, and ammonium nitrogen at the beginning and end of composting. Since the conductivity of the A3 treatment exceeded the safety threshold for plant growth, subsequent experiments were conducted on the remaining three treatments. From Table 2, we can see that at the end of composting, the ammonium nitrogen content had decreased, and the nitrate nitrogen content had increased in all three treatments compared to the initial levels, aligning with prior research [33]. While the differences in ammonium nitrogen content among the three treatments were not significant at the outset, after composting, the ammonium nitrogen content of all three treatments decreased to varying extents, likely attributable to the metabolic activities of microorganisms converting ammonium nitrogen to nitrate nitrogen during composting [34]. The nitrate nitrogen contents of the three treatments were significantly different at the beginning and increased by the end of composting. The A1 treatment exhibited the highest nitrate nitrogen content, significantly differing from both the CK and A2 treatments. By the end of composting, the readily available phosphorus content of all three treatments had increased compared to the initial levels. This may be due to the decomposition of organic-bound phosphorus by microorganisms during composting, resulting in the conversion of phosphorus into inorganic forms that plants can absorb, thereby raising the readily available phosphorus content. The A1 treatment had the highest readily available phosphorus content at the end of composting, which was 261% higher than the initial level and significantly different from both the CK and A2 treatments. Furthermore, the readily available potassium content of the A1 and A2 treatments had increased compared to their initial levels, potentially due to the release of potassium ions from the decomposition of potassium-rich organic materials during composting.

Total nitrogen content was monitored throughout the study (Figure 4a), and it was observed that the nitrogen content of A2 gradually decreased during the first three days and then began to increase, while the nitrogen content of CK and A1 steadily declined during the first six days before increasing. This could be caused by microorganisms, which accelerated the temperature rise and caused the volatilization of TN in the form of ammonia during the warming stage [35]. At the end of composting, the nitrogen content of the A1 treatment reached 33.147 g/kg, significantly different from the CK and A2 treatments. Ultimately, the C/N ratio of all three treatments decreased to varying extents, consistent with previous studies [36].

### 3.4. Changes in Organic Matter and Humic Acid Content

Organic matter serves as the primary source of soil nutrients [37]. As can be seen from Table 3, in this study, the organic matter content in all three treatments decreased during the composting process, particularly in treatment A1, where the organic matter content experienced a 16.42% reduction by the end of composting compared to the initial value. This reduction resulted from the decomposition of organic carbon resources into CO_2_ and H_2_O by active thermophilic microorganisms during the composting process, leading to the decomposition of a substantial amount of total organic carbon (TOC) in the compost [38]. Throughout the composting process, microorganisms degrade putrescible organic matter, releasing heat, reducing water content and pathogens, and generating precursors of humic substances (HS). Fungi, particularly those belonging to the genera Trichoderma, Aspergillus, and Penicillium, play a crucial role in the degradation of recalcitrant organic matter. These fungi possess the ability to produce a variety of specific enzymes, including ligninase and cellulase, which facilitate the degradation of complex organic matter such as lignocellulose and promote the formation of humic substances. This study highlights the significance of these fungi and their enzymes in the degradation of recalcitrant organic matter while investigating the processes involved in lignocellulosic decomposition. As the most crucial product of the composting process, HS plays a significant role in evaluating composting efficiency [39]. The production of HS with high aromaticity and polymerization in compost is vital for chelating metals, decreasing phytotoxicity, and assembling organic nutrients [40]. Consequently, compost rich in mature HS is widely regarded as a promising approach for soil remediation. Humic substances constitute a component of total humus carbon, which represents the total amount of carbon elements in humic substances. Therefore, the content and quality of humic substances directly influence the magnitude of total humus carbon. Moreover, the degree of aggregation and chemical structure of humic substances also affect the magnitude of total humus carbon. In this study, the total humus carbon amount increased by varying degrees in all three treatments, especially in the A1 treatment, which concluded with a 22.3% increase compared to the initial value. Although a clear correlation between total humus carbon and humic acid carbon has not been established in the literature, the increase in total humus carbon in this study primarily manifested as a change in humic acid carbon. The pre- and post-composting humic acid carbon amounts varied considerably in the three treatments, whereas the changes in fulvic acid carbon and humin carbon amounts were relatively minor in comparison.

### 3.5. Changes in Microbial Communities

Composting primarily depends on the interaction between microbial activities and diverse microbial communities. The composition and succession of microbial communities are intimately related to the maturity and quality of compost [41].

Figure 5a illustrates the changes in bacterial phyla and genera before and after the three treatments. In this figure, the top four bacterial phyla and the top twenty-nine bacterial genera, accounting for the highest percentage in the pile before and after composting, were selected. In accordance with most studies, the phyla Firmicutes, Bacteroidota, Proteobacteria, and Actinobacteriota were the most dominant bacterial phyla in the composting process [42]. In this study, Proteobacteria and Actinobacteriota constituted the primary flora in the pile, with the abundance of Proteobacteria decreasing and that of Actinobacteriota increasing before and after the three treatments. Proteobacteria can decompose and degrade various harmful substances, avoiding pollution of crops and the environment. Meanwhile, Actinobacteriota facilitate the decomposition and transformation of organic matter, enhancing the quality of compost through flora succession [31]. At the genus level, composted piles displayed a large number of beneficial bacteria, commonly found in the water column and soil. Among them, the relative abundance of two bacterial genera, *Streptomyces* and *Thermostaphylospora*, was the highest. *Streptomyces* and *Thermostaphylospora* possess favorable soil amelioration effects, improve soil fertility and drought resistance, and maintain soil health and ecological balance [43,44].

Figure 5b presents the changes in fungal phyla and genera before and after the three treatments. The results revealed that before composting, members of the Basidiomycota phylum constituted 83.88%, 87.68%, and 89.23% of the CK, A1, and A2 treatment piles, respectively. After composting, members of the Ascomycota phylum prevailed, constituting 83.29%, 99.16%, and 95.14% of the F, G, and H treatment piles, respectively. The phylum Basidiomycota and Ascomycota are widely distributed in the composting process, and their dominance has been documented [45,46]. At the genus level, the fungal species in the piles were primarily *Auricularia* spp. before composting, representing 83.35%, 86.67%, and 88.27% of the fungal species in the CK, A1, and A2 treatment piles, respectively. However, these species had vanished entirely by the end of composting. The three genera *Talaromyces*, *Melanocarpus*, and *Thermomyces*, which were less abundant in the pre-composting period, exhibited a significant increase in abundance after composting, particularly the genus *Thermomyces*, which accounted for 64.64%, 83.54%, and 47.07% of the fungal species in the F, G, and H piles, respectively, at the end of composting. In the A1 treatment, the most substantial changes were observed in the genus *Thermomyces*. All of these genera can decompose organic matter and promote organic matter cycling in the soil. Additionally, they can produce beneficial enzymes, such as cellulases and xylanases, which assist plants in the soil in better absorbing nutrients [47].

### 3.6. Correlation between Environment Factors and Microbial Community

Previous studies have shown that environmental factors are directly or indirectly related to microorganisms in the composting process [48]. Figure 6a illustrates the effect of environmental factors on the microbial community, as analyzed by canonical correspondence analysis (CCA). CCA1 and CCA2 accounted for 61.45% and 10.02% of the variation in microbial community structure, properly, indicating that the selected environmental parameters explained most of the variation. *Raoultella* and *Auricularia* were positively correlated with NH_4_^+^ and negatively correlated with NO_3_^−^, pH, EC, and total humus carbon. This suggests that the two microorganisms with the largest proportion in the pre-compost pile were gradually eliminated as nitrifying bacteria and physicochemical properties in the pile changed and did not adapt to the weakly alkaline and high electrical conductivity environment. Conversely, the other six microorganisms were positively correlated with NO_3_^−^, pH, EC, and total humus carbon. indicating that they play a significant role in altering these physicochemical properties.

In Figure 6b, CCA1 and CCA2 accounted for 62.21% and 10.76%, respectively. Lignin, cellulose, and hemicellulose were positively correlated with *Raoultella* and *Auricularia* and negatively correlated with the other six microorganisms. This reveal that the degradation of lignocellulose was not related to *Raoultella* and *Auricularia*, in the time, the other six microorganisms influenced the degradation of lignocellulose. The status of bacterial and fungal communities can greatly affect the degradation and conversion of lignocellulose during the composting process. This demonstrates that the improvement of functional microorganisms can regulate environmental factors, providing exceptional strategies to improve composting efficiency.

## 4. Conclusions

In conclusion, this study demonstrates the impact of different proportions of straw on the physicochemical properties of compost generated from the spent mushroom substrate and chicken manure produced during the cultivation of mushrooms. The findings indicate that the waste mushroom residue generated from the cultivation with 20% straw added outperforms other treatments in terms of physicochemical properties and microbial diversity. Although there are some limitations in the study, such as the only approximate calculation of straw addition amounts, the results provide valuable insights for improving resource recycling rates. Future research could further optimize the amount of straw added, which may lead to a more comprehensive understanding of the composting process. These findings offer a more environmentally friendly and efficient waste disposal method for the mushroom industry, thereby reducing the industry’s impact on the environment and increasing resource recycling rates.

## Figures and Tables

**Figure 1 jof-09-00925-f001:**
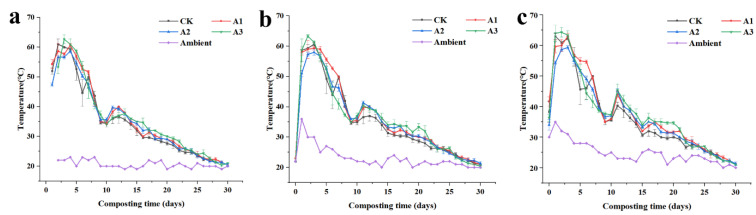
Effect of different straw additions on the temperature during composting. (**a**): The temperature at 8:00; (**b**): The temperature at 14:00; (**c**): The temperature at 20:00. CK: No straw added; A1: 20% straw added; A2: 40% straw added; A3: 80% straw added.

**Figure 2 jof-09-00925-f002:**
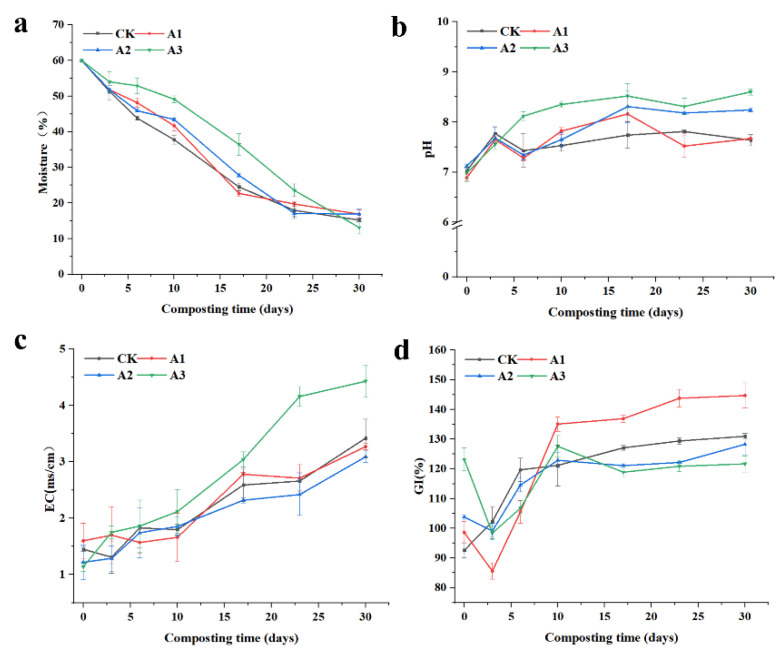
Effect of different straw additions on GI, pH, EC, and moisture during composting. (**a**): Changes in moisture; (**b**): Changes in pH; (**c**): Changes in EC; (**d**): Changes in GI. CK: No straw added; A1: 20% straw added; A2: 40% straw added; A3: 80% straw added.

**Figure 3 jof-09-00925-f003:**
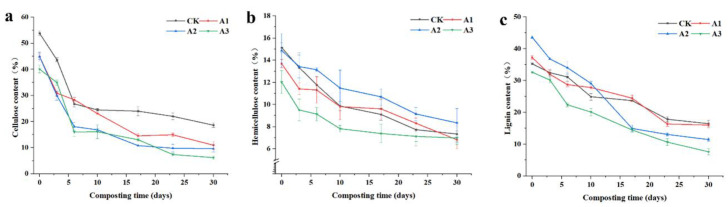
Effect of different straw additions on cellulose, hemicellulose, and lignin during composting. (**a**): Changes in cellulose; (**b**): Changes in hemicellulose; (**c**): Changes in lignin. CK: No straw added; A1: 20% straw added; A2: 40% straw added; A3: 80% straw added.

**Figure 4 jof-09-00925-f004:**
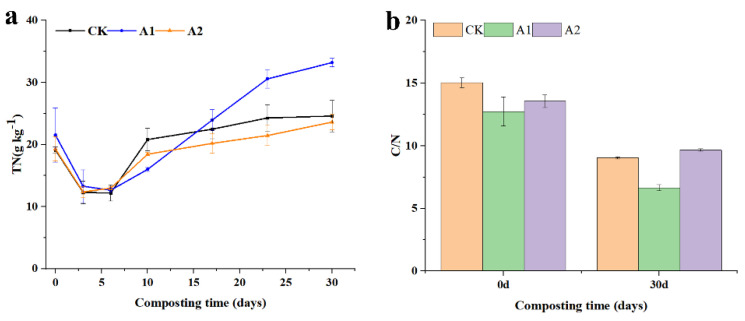
Effect of different straw additions on TN and C/N during composting. (**a**): Changes in TN; (**b**): Changes in C/N. CK: No straw added; A1: 20% straw added; A2: 40% straw added.

**Figure 5 jof-09-00925-f005:**
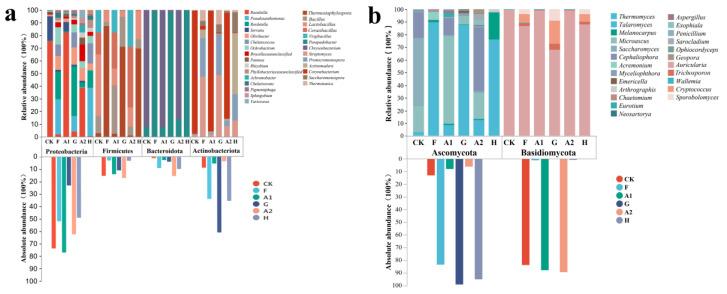
Effect of different straw additions on the bacterial and fungal communities before and after composting. (**a**): Changes in bacterial communities; (**b**): Changes in fungal communities. CK: no straw added; A1: 20% straw added; A2: 40% straw added; F: after CK composting; G: after A1 composting; H: after A2 composting.

**Figure 6 jof-09-00925-f006:**
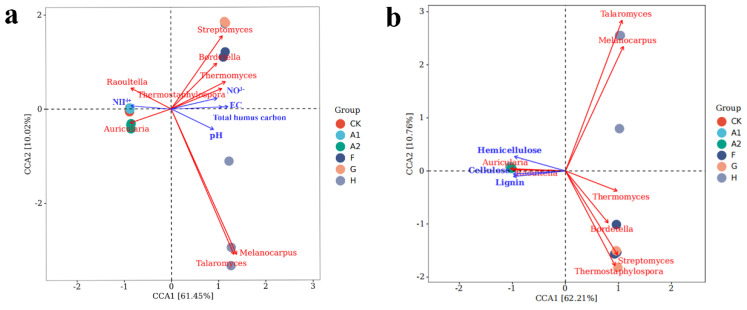
Canonical correspondence analysis of the correlation between microbial communities and environmental factors; (**a**): NH_4_^+^, NO_3_^−^, pH, EC, and total humus carbon; (**b**): Lignin, cellulose, and hemicellulose. CK: no straw added; A1: 20% straw added; A2: 40% straw added; F: after CK composting; G: after A1 composting; H: after A2 composting.

**Table 1 jof-09-00925-t001:** Proportions of straw and wood chips in the test materials and the carbon to nitrogen ratio.

Treatment	Sawdust	Straw	Wheat Bran	Soybean Meal	Lime	Light Calcium Carbonate	Chicken Manure	Percentage of Carbon Source Content	Percentage of Nitrogen Source Content	Carbon/Nitrogen Ratio
CK	80%	0%	15%	3%	1%	1%	One-third of the volume of the residue	28.60	1.90	15.05
A1	60%	20%	15%	3%	1%	1%		26.70	2.14	12.47
A2	40%	40%	15%	3%	1%	1%		27.81	2.04	13.63
A3	0%	80%	15%	3%	1%	1%		25.91	1.85	14.00

**Table 2 jof-09-00925-t002:** Content of Available-P, Available-K, NO_3_^−^-N, and NH_4_^+^-N before and after different treatments of composting.

Treatment	Available-P (g/kg)		Available-K (g/kg)		NO_3_^−^-N (mg/kg)		NH_4_^+^-N (g/kg)	
	Before	After	Before	After	Before	After	Before	After
CK	0.48 ± 0.02 b	1.21 ± 0.02 b	9.17 ± 0.88 a	7.34 ± 0.19 b	827.17 ± 3.18 b	1287.16 ± 9.23 b	2.02 ± 0.12 a	1.35 ± 0.06 b
A1	0.44 ± 0.02 c	1.59 ± 0.03 a	8.04 ± 0.22 b	9.16 ± 0.23 a	638.04 ± 4.07 c	1597.50 ± 8.79 a	2.14 ± 0.31 a	1.77 ± 0.01 a
A2	0.65 ± 0.01 a	1.55 ± 0.02 a	4.85 ± 0.04 c	7.85 ± 0.35 b	888.65 ± 13.74 a	949.90 ± 11.71 c	2.36 ± 0.08 a	1.20 ± 0.01 c

CK: no straw added; A1: 20% straw added; A2: 40% straw added. Note: Those with different letters in the same column showed significant differences among the groups (*p* < 0.05).

**Table 3 jof-09-00925-t003:** Total carbon content of organic matter and humus before and after different treatments of composting.

Treatment	Organic Matter (g/kg)		Total Humus Carbon (g/kg)		Humic Acid Carbon (g/kg)		Fulvic Acid Carbon (g/kg)		Humin Carbon (g/kg)	
	Before	After	Before	After	Before	After	Before	After	Before	After
CK	438.72 ± 2.04 a	404.15 ± 2.99 a	314.49 ± 1.44 a	344.59 ± 0.90 c	26.95 ± 0.28 b	56.40 ± 0.49 b	49.12 ± 0.22 a	55.69 ± 1.40 a	238.62 ± 2.25 b	239.52 ± 0.30 b
A1	438.91 ± 1.67 a	370.02 ± 7.98 b	309.86 ± 1.88 b	378.99 ± 0.37 a	19.66 ± 0.18 c	79.90 ± 0.28 a	42.68 ± 1.22 b	50.96 ± 2.80 b	248.73 ± 1.40 a	205.99 ± 1.84 c
A2	428.85 ± 3.85 b	372.37 ± 2.69 b	309.67 ± 1.21 b	360.22 ± 1.00 b	32.04 ± 2.25 a	79.47 ± 1.04 a	26.98 ± 1.82 c	28.36 ± 0.94 c	251.83 ± 0.78 a	253.04 ± 1.51 a

CK: no straw added; A1: 20% straw added; A2: 40% straw added. Note: Those with different letters in the same column showed significant differences among the groups (*p* < 0.05).

## Data Availability

Data will be made available on request.

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
