# Peer review of "Optimizing Straw-Rotting Cultivation for Sustainable Edible Mushroom Production: Composting Spent Mushroom Substrate with Straw Additions"

_jof, 2023, doi:10.3390/jof9090925_

Round 1
Reviewer 1 Report
Dear Editor,
I am here with submitting of the review manuscript ID jof-2593004 entitled "Optimizing Straw-Rotting Cultivation for Sustainable Edible Mushroom Production: Composting Spent Mushroom Substrate with Straw Additions".
The authors Ma et al. presented haw the waste mushroom residue generated from the cultivation with 20% straw addition outperform other treatments in terms of physicochemical properties and microbial diversity and following the natural rules and maybe in future translating those into the business model apply circularity,
My suggestion is to make minor changes
Lines 28 – name some of many reports after 2014
Lines 272 – name some of many reports after 2013
- In Table 1. the formulation of Yu Muer residue combined with other straw additions - Find out what the abbreviations mean
In my opinion, the manuscript is well written, the data presented in this are interesting, and my suggestion is to accept it for publication in the Journal of Fungi
Author Response
Dear reviewer,
Thank you very much for your valuable comments and suggestions on my paper. Based on your feedback, I have made the following changes:
1. for the suggestion "Lines 28 - name some of many reports after 2014", I have added in line 28 what I consider to be important reports on edible mushroom production published after 2014.
2. for the suggestion "Lines 272 - name some of many reports after 2013", I have added references to lignocellulose degradation by fungi secreting lignocellulase after 2013 in line 272.
3. In "Table 1. the formulation of Yu Muer residue combined with other straw additions - Find out what the abbreviations mean". I have consulted the relevant information and have labelled the full names of the abbreviations used in Table 1.
I hope that the above changes meet your requirements. Thank you again for your careful review and helpful suggestions. If you have any other suggestions or need more information, I will be more than happy to revise and add them again.
Best wishes.
Changtian Li
Reviewer 2 Report
In recent years, the world of mushroom cultivation has faced new challenges related to the optimization of recipes causing rotting of straw and the management of used mushroom substrate.
The results of the study underscore the importance of including straw in mushroom cultivation as a means to improve the subsequent composting of the spent mushroom substrate. This approach is not only compatible with sustainable agricultural practises, but also offers a promising route to environmental protection and profitable mushroom production.
The manuscript has been meticulously prepared with great attention to detail. The methodology is presented in a clear and concise way. The graphical elements are accurately represent the data.
In the study under consideration, the optimization of preparations causing straw rot in mushroom cultivation and the management of used mushroom substrate were investigated.
The inclusion of straw in mushroom cultivation is effective from an environmental protection point of view, but the specificity of these environmental benefits has not been investigated. What are the environmental impacts and benefits of this approach and how does it compare to other waste management methods?
What are the real savings related to the inclusion of straw in mushroom cultivation and subsequent composting?
Author Response
Dear reviewer.
Q: Incorporating straw into mushroom cultivation is effective from an environmental protection point of view, but the specificity of these environmental benefits has not been studied. What are the environmental impacts and benefits of this approach?
A: From the point of view of environmental protection, the use of crop straw for mushroom cultivation has the following significant advantages:
a. Resource recycling: Crop residues, as abundant agricultural wastes, can be utilised in mushroom cultivation to maximise the usefulness of resources and save the consumption of new resources.
b. Reducing environmental pollution: In most cases, burning straw randomly will cause serious air pollution. However, using straw in mushroom production can effectively prevent such pollution and reduce air pollution.
c. Improve soil fertility: Waste mushroom substrate after a growing cycle is a good organic fertiliser, which can improve soil structure and soil fertility.
Q: How does it compare to other waste management methods?
A: Compared with other waste disposal methods such as landfill and incineration, growing mushrooms with straw not only makes effective use of waste, but also provides valuable products and nutrient-rich fertilisers, creating a virtuous cycle.
Q: What are the real savings of adding straw to mushroom growing and subsequent composting?
The real savings of using straw to grow mushrooms are mainly due to
a. Cost Savings: Using straw is less expensive than commercial mushroom media, which reduces cost of production.
b. Ecological benefits: Using straw is favourable to environmental protection and indirectly produces ecological benefits, such as relieving environmental pressure and improving air quality.
Best regards,
Changtian Li